Gas chromatography (GC) fingerprinting and immunomodulatory activity of polysaccharide from the rhizome of Menispermum dauricum DC

Yang Pei 1
Zhai Yang 2
Ma Yan 1
Mao Beibei 1
Wang Fengshan 3
Li Li 4
Luan Lijuan 2 15553115573@163.com
Liu Yuhong 1 yhliu@sdutcm.edu.cn
1 School of Pharmaceutical Sciences, Shandong University of Traditional Chinese Medicine , Jinan , China
2 Shandong Cancer Hospital and Institute, Shandong First Medical University and Shandong Academy of Medical Sciences , Jinan , China
3 School of Pharmaceutical Sciences, Shandong University, NMPA Key Laboratory for Quality Research and Evaluation of Carbohydrate-based Medicine, National Glycoengineering Research Center , Jinan , China
4 Sishui Siheyuan Culture and Tourism Development Company, Ltd , Sishui , China
Mora-Montes Héctor
Electronic publication date: 2022 Aug 22
Publication date: 2022
Volume: 10
Electronic Location ID: e13946
Received 2022 Jul 4; Accepted 2022 Aug 3
Copyright: © 2022 Yang et al.
Copyright year: 2022
Copyright holder: Yang et al.
License: This is an open access article distributed under the terms of the Creative Commons Attribution License, which permits unrestricted use, distribution, reproduction and adaptation in any medium and for any purpose provided that it is properly attributed. For attribution, the original author(s), title, publication source (PeerJ) and either DOI or URL of the article must be cited.
License URL: https://creativecommons.org/licenses/by/4.0/

Keywords: Menispermum dauricum DC, GC fingerprinting, Immunomodulatory activity, TLR4-MAPKs/NFκB

Funding: National Natural Science Foundation of China 81973218 Taishan Industry Leading Talents Project tscy20200410 Technology Development Program of TCM of Shandong Province 2019-0024 Open Projects Fund of NMPA Key Laboratory for Quality Research and Evaluation of Carbohydrate-based Medicine 2021QRECM02 Shandong Key Laboratory of Carbohydrate Chemistry and Glycobiology, Shandong University 2021CCG05 We received support in the form of grants from the National Natural Science Foundation of China (81973218), the Taishan Industry Leading Talents Project (tscy20200410), the Technology Development Program of TCM of Shandong Province (2019-0024), the Open Projects Fund of NMPA Key Laboratory for Quality Research and Evaluation of Carbohydrate-based Medicine (No. 2021QRECM02) and the Shandong Key Laboratory of Carbohydrate Chemistry and Glycobiology, Shandong University (2021CCG05). The funders had no role in study design, data collection and analysis, decision to publish, or preparation of the manuscript.

==============================
This research aimed to establish the gas chromatography (GC) fingerprints and examine the immunomodulatory activity of the rhizome of Menispermum dauricum polysaccharides. In this study, the preparation conditions were optimized by the response surface method (RSM). GC is an effective and sensitive technique employed to measure the composition of monosaccharides; the GC fingerprints of total polysaccharides from 10 batches of the rhizome of M. dauricum (tMDP) were established, and chemometrics methods were adopted to examine the differences and similarities of tMDP from distinct regions. The similarity evaluation illustrated that the polysaccharides derived from the rhizome of M. dauricum from different origins were highly similar. The results of principal components analysis (PCA) illustrated that all the tMDPs may be integrated into one group within the 95% confidence interval, but the rhizome of M. dauricum from different origins could also be distinguished in the plot of PCA scores. Then, the major bioactive fraction MDP was purified and obtained by column chromatography. Our previous study showed that MDP exhibited significant immunomodulatory activity, but the mechanism of the in vitro immunomodulatory activity of MDP is unclear. The macrophage activation induced by MDP was abolished when Toll-like receptor 4 (TLR4) signaling was knocked down by the TLR4 inhibitor. Furthermore, western blot analysis illustrated that MDP activated RAW264.7 cells through MAPKs and NFκB pathways induced by TLR4. This research offers a theoretical foundation for quality control and additional study as a potential immunomodulator of MDP.

Introduction

The rhizome of Menispermum dauricum DC (Menispermaceae), referred to as Bei Dou Gen in Chinese, which is mainly produced in northeastern, northern, and eastern China, is a traditional Chinese medicine herb that has had widespread application in modern medicine to treat diarrhea, sore throats, colitis, and rheumatic arthralgia, as well as other conditions (Wu & Jin, 2007). It contains various chemical components such as alkaloids, polysaccharides, quinones, cardiac glycosides, lactones, saponins, tannins, proteins, and resins (Wu et al., 2018; Zhou et al., 2018). The M. dauricum rhizome’s main chemical components are alkaloids. These alkaloids exert a variety of biological properties, including those that influence the cardiovascular system, antitumor activity, and antiarrhythmic function. Additionally, the injection of total alkaloids has been used therapeutically for a considerable period for the treatment of chronic tracheitis, as well as arthralgia and throat sores (Chen et al., 2012).

The polysaccharides are also significant bioactive constituents of the rhizome of M. dauricum, which exhibited numerous pharmacological properties while having no toxic effects (Li et al., 2006; Liang et al., 2005; Lin et al., 2013a, 2013b). In most cases, the quality control of polysaccharides is primarily dependent on the phenol sulfuric acid technique, which involves establishing the content of sugar present (Zhang et al., 2021). The phenol sulfuric acid approach, on the other hand, is incapable of providing any structural properties of polysaccharides because of its low level of specificity. Additionally, research suggested that adulterated or low-quality products cannot be identified in an efficient manner (Li et al., 2019). Moreover, the biological properties of polysaccharides are based on their numerous morphological characteristics, such as the composition of their monosaccharides, the kind of glycosidic bonds, and the distribution of their molecular weight (Li et al., 2019). In particular, the examination of the polysaccharide’s monosaccharide composition is the first and initial phase in the analysis of polysaccharides, which needs to be taken into consideration to set up a quality control system for polysaccharides and products associated with them.

Recently, the use of fingerprint profiling as a technology and methodology for quality control of compounds generated from plants has gained widespread recognition as a practical and effective technique (Li et al., 2019). In addition, chemometrics is a robust tool for analyzing fingerprints premised on chemical data, such as principal component analysis (PCA) and similarity analysis (SA), thus contributing to the analysis of the raw data gathered from fingerprints (Dong et al., 2020; Liu et al., 2015a). Fingerprinting methods have been utilized effectively to control quality as well as standardize plant polysaccharides, including Panax, Cordyceps, Ganoderma, tea, and Lycium barbarum polysaccharides (Sun et al., 2014).

Polysaccharides derived from natural sources have garnered a lot of interest in recent years because they do not have any toxic effects in studies to date and have potent immunomodulatory properties (Pan et al., 2021; Zhong et al., 2021; Zhou et al., 2021). As a result, there is a remarkable demand for the extraction and purification of new immunomodulatory polysaccharides derived from natural products that are both safe and efficacious. Macrophages, which are powerful phagocytic cells, are present in nearly all of the body’s tissues. Macrophage activation is an integral process in both adaptive and innate immune responses, which may trigger and propagate defense responses against infections (Ma et al., 2019; Wang et al., 2017b). Some botanical polysaccharides could be recognized and combined with specific receptors on macrophages to enhance the viability of macrophages against pathogenic microorganisms and tumorigenesis by the promotion of phagocytic and the cytokines of nitric oxide (NO), interleukin (IL)-6 and tumor necrosis factor (TNF)-a (Yu et al., 2012). The polysaccharides with immunomodulatory action that are derived from plant products have no reports of allergies or adverse effects and have been believed to be potential candidates for use as immunoregulators (Zhao et al., 2020).

In this research, the fingerprint of the rhizome of M. dauricum polysaccharide was established and analyzed using gas chromatography (GC) and chemometrics, which offered a technique that was both reliable and effective for controlling the quality of the rhizome of M. dauricum and provided the groundwork for the future expansion of its functional products. Then, the major bioactive fraction MDP was obtained by column chromatography. Moreover, the in vitro immunomodulatory activities and the mechanism of MDP on macrophages were also evaluated.

Materials and Methods

Materials

The rhizome of M. dauricum plants for each of the 10 batches, denoted by the letters S1 to S10, was gathered from various regions throughout China (Table 1). The voucher samples of rhizome of M. dauricum were deposited in the School of Pharmaceutical Sciences of Shandong University of Traditional Chinese Medicine, Jinan, China.

Table 1 The source, neutral sugar content and principal component analysis composite score of the total polysaccharides from the rhizome of M. dauricum.

No.	Source	Neutral sugar content (%)	Principal component analysis composite score	
S1	Shandong	20.34	−1.777427	
S2	Shandong	19.54	0.0950919	
S3	Heilongjiang	20.32	0.8092878	
S4	Shandong	20.77	−1.506477	
S5	Heilongjiang	19.47	−0.156665	
S6	Liaoning	22.77	0.6893036	
S7	Jilin	20.12	0.5907167	
S8	Shandong	21.91	0.1607214	
S9	Liaoning	22.61	−1.386776	
S10	Jilin	20.86	2.4822341	

Chemicals: The DEAE-cellulose-52 was procured from Yuanye Biological Co., Ltd. (Shanghai, China). Sephadex G-50 and Sephacryl S-100 were procured from GE Healthcare Life Sciences (Piscataway, NJ, USA). Guoyao Group Co., Ltd. (Beijing, China) provided glucose and galactose. Arabinose, fucose, mannose, and rhamnose were all supplied by Macklin Biochemical Technology Co., Ltd. (Shanghai, China). Lipopolysaccharide (LPS) was procured from Sigma-Aldrich (St. Louis, MO, USA). The ELISA kits employed in the IL-6, NO, TNF-α, and IgM tests were retrieved from Enzyme-linked Biotechnology Co., Ltd. (Shanghai, China). Antibodies against MyD88, NFκB, JNK, ERK, and P38, as well as GAPDH, were procured from ABclonal (Wuhan, China) while other antibodies were procured from Cell Signaling Technology (Beverly, MA, USA).

Experimental design of RSM

The independent parameters chosen were the liquid-solid ratio (A, mL/g), extraction time (B, min) and extraction temperature (C, °C) whereas the dependent parameter was the polysaccharide extraction yield, premised on the findings from the single-factor experiments. The computation for the extraction yield was performed based on the formula below:

Extraction yield (%) = [crude polysaccharide weight (g)/rhizome of M. dauricum weight (g)] × 100%.

A three-factor three-level response surface optimization experiment and prediction model were constructed with the aid of the Design-Expert software (version: 8.0.5).

GC fingerprint analysis

Preparation of polysaccharide extracts

Distilled water was used in the process of extracting crude polysaccharides from the rhizome of M. dauricum under the optimal extraction parameters, which were determined through response surface optimization. Following the completion of three rounds of extraction, the aqueous extract was subjected to a vacuum and concentrated. After that, four times the volume of ethanol was introduced into the solution to precipitate the polysaccharide, and afterward, the mixture was left to stand at 4 °C for an entire night. The precipitate was collected and deproteinized by means of the Trichloroacetic acid (TCA)-n-butanol method and then subjected to freeze-drying to produce a total polysaccharide fraction (tMDP) (Yang et al., 2022).

Complete acid hydrolysis

Hydrolysis of 10 mg of tMDP was performed using 3 mL of 2 mol/L trifluoroacetic acid (TFA) in a vial that was sealed at 110 °C for 6 h. To get rid of the TFA residue, the mixture was combined with 1 mL of methanol, and then it was rotary-evaporated thrice under a vacuum until it was completely dry. The hydrolyzed polysaccharide samples were prepared for derivatization (Liu et al., 2015b).

Preparation of derivatization and GC analysis

The GC method was employed to measure the composition of the monosaccharides. The hydrolyzed tMDP and monosaccharide standards were added with hydroxylamine hydrochloride, pyridine, and inositol hexaacetate, and stirred at 90 °C for 30 min. Then, acetic anhydride was introduced and the reaction continued for 30 min at 90 °C for acetylation. A GC examination was performed on the products of the reaction. The temperature program was set to 170 °C for 3 min, 170 °C to 178 °C at a rate of 0.5 °C/mins for 3 min, and elevated to 210 °C for 5 min at a rate of 2 °C/mins (Yang et al., 2022).

Extraction and purification of polysaccharides

tMDP was extracted from the S10 batch (highest PCA score) of the rhizome of M. dauricum by the method above, and the TCA-n-butanol method was used to remove protein. Subsequently, the purification of tMDP was done utilizing the Sephadex G-50 column (1.75 cm × 66 cm), Sephacryl S-100 column (1.75 cm × 66 cm), and DEAE-52 cellulose column (5.5 cm × 30 cm), and A white purified polysaccharide (MDP) was obtained by collecting the major fraction and lyophilizing it (Yang et al., 2022).

Immunomodulatory activity in vitro

Cell culture

Beina Chuanglian Biotechnology Co., Ltd. (Beijing, China) provided the RAW264.7 cells utilized in this study. At a temperature of 37 °C, the cells were incubated in DMEM that had been supplemented with streptomycin sulfate (100 g/ml), penicillin (100 units/mL), and 10% fetal bovine serum. The environment of the incubator was humidified and contained 5% carbon dioxide.

Cell viability test

By conducting the Methyl thiazolyl tetrazolium (MTT) test in vitro, we examined the effect that varying concentrations of MDP had on the RAW264.7 cells’ viability. In general, 96-well microplates were seeded with RAW264.7 cells at a density of 1 × 104 cells per well. After the cells had been cultured at 37 °C for 24 h with 5% carbon dioxide, they underwent a second round of incubation at 37 °C for another 24 h with MDP samples at increasing dosages (0, 10, 50, 100, 200, and 400 µg/mL) or LPS (1 µg/mL, positive control). Following the initial incubation, 20 μL of the MTT solution with a concentration of 5 mg/mL was introduced into each well, and the plates were subsequently re-incubated at 37 °C for a further 4 h in the medium. After that, the medium was carefully aspirated, and each well was then treated using 150 μL of DMSO so that the dissolving crystals could be dissolved. At last, to ascertain the absorbance at 570 nm, we made use of a microplate reader (BioTek Instruments Inc., Winooski, VT, USA) (Wang et al., 2017a).

Cell viability % = (OD treatment group)/(OD control group) × 100%.

TLR4 pathway blocking experiment

RAW264.7 cells were seeded at a density of 1 × 104 cells/well into 96-well microplates for 24 h. Following the adhesion of the cells, various doses of MDP (50, 100, and 200 μg/mL), as well as LPS, were introduced, and TAK-242 was added at the same time to make the final concentration of 20 μg/mL. Next, the concentrations of IL-6, TNF-α, and NO in the supernatant were detected with the aid of ELISA kits (ML BIO Biotechnology, Shanghai, China) that are commercially available. The assays were carried out in compliance with the guidelines that were supplied by the manufacturer, after which the cytokine levels were calculated using the standard curves.

Western blotting analysis

After cultivating the RAW 264.7 cells for 24 h on a tissue culture plate with six wells having a flat bottom at 5 × 106 cells per well, the MDP was added at a final concentration of 50 μg/mL in order to activate these cells for 24 h. The positive control consisted of the cells that had been treated using LPS at a concentration of 1 μg/mL. At the end of the incubation period, all of the conditioned cells were harvested and processed to produce the total proteins. Briefly, samples were prepared by RIPA (radioimmune precipitation assay) with protease and phosphatase inhibitors. The BCA technique was used to determine the protein. After denaturation, 30 μg of protein per well was deposited via electrophoresis onto SDS-PAGE and then transferred onto PVDF membranes. 10% nonfat milk (prepared in TBS supplemented with 0.1% Tween 20) was employed in blocking the membranes at ambient temperature for 4 h. Afterward, the membranes were probed with primary antibodies against P38 (1:750), p-P38 (1:1,000), ERK (1:1,500), p-ERK (1:1,500), JNK (1:1,500), p-JNK (1:2,000), NFκB (1:1,500), p-NFκB (1:1,000), MyD88 (1:1,500), TLR4 (1:1,000), as per the guidelines stipulated by the manufacturer. After being rinsed in TBST, the membranes were subjected to incubation with the secondary antibody (1:5,000) for 1 h at ambient temperature. ECL chemiluminescence detection kit (Millipore, MA, USA) was utilized to analyze the signals, which were subsequently quantified with an Amersham Imager 600 Chemiluminescence imaging system (GE, Boston, MA, USA).

Results

Extraction conditions optimization

To maximize the extraction rate of polysaccharides and herb utilization, and to lay the foundation for the subsequent fingerprinting study, RSM was adopted to generate optimal conditions for extracting the rhizome of M. dauricum polysaccharides. Premised on the findings obtained from the single-factor experimental analysis Figs. 1A–1C, a 3-factor 3-level response surface experiment was carried out for optimizing the extraction conditions of polysaccharides. To perform the variance analysis, we utilized the Design-Expert software (Table 2) to identify an optimal region for the study by fitting the mathematical model with the experimental data.

Figure 1 Influences of extraction variables on the polysaccharides yield from the rhizome of M. dauricum; response surface plots and contour plots.

(A) Effect exerted by the liquid/solid ratio (mL/g) on polysaccharides yield from the rhizome of M. dauricum; (B) Effect exerted by the extraction time (min) on polysaccharides yield from the rhizome of M. dauricum; (C) Effect exerted by the extraction temperature (°C) on polysaccharides yield from the rhizome of M. dauricum; (D) plots of response surface presenting the interactive effects of liquid/solid ratio (a) and extraction time (b); (E) response surface plots presenting the interactive effects of liquid/solid ratio (a) and extraction temperature (c); (F) response surface plots presenting the interactive effects of extraction duration (b) and extraction temperature (c); (G) contour plots presenting the interactive effects of A and B; (H) contour plots presenting the effects of the interaction of a and c; (I) contour plots presenting the interactive effects of b and c. The values are presented as the mean ± SD, n = 3.

Table 2 Analysis of variance in the regression model.

Source	Square sum	df	Mean square	F value	P value	Significance	
Model	0.69	9	0.077	357.11	<0.0001	**	
A-A	0.0072	1	0.0072	33.60	0.0007	**	
B-B	0.00845	1	0.00845	39.43	0.0004	**	
C-C	0.029	1	0.029	134.4	<0.0001	**	
AB	0.000225	1	0.000225	1.05	0.3396		
AC	0.000625	1	0.000625	2.92	0.1314		
BC	0.000225	1	0.000225	1.05	0.3396		
A2	0.43	1	0.43	1,996.38	<0.0001	**	
B2	0.13	1	0.13	593.19	<0.0001	**	
C2	0.037	1	0.037	172.7	<0.0001	**	
Residual	0.00150	7	0.0002143				
Lack of fit	0.0009	3	0.0003	2.0	0.2564		
Pure error	0.0006	4	0.00015				
Cor total	0.69	16					
Note:

** P < 0.01 significant different.

To show the type of interaction between variables, 2D contour and 3D response surface plots are presented in Figs. 1D–1I. According to the plots, the optimum extraction conditions for polysaccharides were a ratio of liquid to the raw material of 19.79 mL/g, an extraction duration of 116.47 min, and a temperature for extraction of 86.86 °C. The maximum anticipated polysaccharides yield, under the optimal conditions, was 1.781%. However, for the sake of the operability and convenience of production, there could be modifications to the experimental parameters: the time for extraction was 116 min, the ratio of liquid to the raw material was 20 mL/g, and the temperature used for extraction was 87 °C.

GC fingerprints analysis

GC fingerprints of tMDP

tMDP was prepared by the optimized extraction conditions of RSM and deproteinization using the TCA-n-butanol method, which was used to establish GC fingerprinting, thus laying the foundation for controlling the quality of the rhizome of M. dauricum. To ascertain the structural characteristics of polysaccharides, it is required and essential to take into account the monosaccharide composition. Compositions of monosaccharides and polysaccharide ratios varied remarkably by the source, which was an essential aspect of the polysaccharide’s biological activity. In the quality control of active polysaccharides throughout the last several years, the monosaccharide composition-associated fingerprints have seen a widespread application. To determine the standard GC fingerprint, GC was utilized to conduct monosaccharide composition analysis on 10 different batches of tMDP. As shown in Fig. 2A, GC fingerprinting of full acid hydrolysates of 10 different batches of tMDP exhibited a high degree of similarity having 11 common peaks. Figure. 2B shows the fingerprint that was used as a reference. The chromatogram of mixed standard monosaccharides depicted in Fig. 2C revealed the presence of seven different components, with peaks 1, 2, 3, 8, 9, 10, and 11 representing rhamnose, arabinose, fucose, mannose, glucose, galactose, and inositol (internal standard), respectively. To sum up the above, tMDP contained six monosaccharides, among which rhamnose, glucose, and galactose were the majority, whereas arabinose, fucose and mannose, were of low content.

Figure 2 GC chromatograms of 10 different batches of tMDP’s complete acid hydrolysates (A), the reference GC fingerprint of tMDP (B), Monosaccharide standards, and tMDP sample (1, Rhamnose; 2, Arabinose; 3, Fucose; 8, Mannose; 9, Glucose; 10, Galactose; 11, Internal standard) (C).

The similarity analysis of the GC fingerprints

Based on the professional program “Similarity Evaluation System for Chromatographic Fingerprint of Traditional Chinese Medicine (TCM)” (Version 2012A), similarity values were computed premised on the GC chromatograms of polysaccharide samples as well as the reference fingerprint. Relative to the reference fingerprint, the similarity across 10 batches of tMDP ranged from 0.943% to 0.998 (Table 3). The great similarity revealed that the previously developed GC fingerprint was particularly suited as one of the multi-dimensional fingerprints for quality control of the rhizomes of M. dauricum.

Table 3 Similarity analysis of 10 batches of tMDP.

No.	S1	S2	S3	S4	S5	S6	S7	S8	S9	S10	R	
S1	1.000	0.984	0.990	0.997	0.990	0.990	0.989	0.973	0.977	0.988	0.995	
S2	0.984	1.000	0.977	0.988	0.977	0.976	0.975	0.998	0.964	0.958	0.987	
S3	0.990	0.977	1.000	0.991	0.998	0.993	0.998	0.968	0.996	0.994	0.998	
S4	0.997	0.988	0.991	1.000	0.990	0.995	0.990	0.977	0.981	0.987	0.997	
S5	0.990	0.977	0.998	0.990	1.000	0.990	0.998	0.970	0.993	0.991	0.997	
S6	0.990	0.976	0.993	0.995	0.990	1.000	0.992	0.963	0.988	0.993	0.996	
S7	0.989	0.975	0.998	0.990	0.998	0.992	1.000	0.967	0.997	0.994	0.997	
S8	0.973	0.998	0.968	0.977	0.970	0.963	0.967	1.000	0.955	0.943	0.979	
S9	0.977	0.964	0.996	0.981	0.993	0.988	0.997	0.955	1.000	0.990	0.992	
S10	0.988	0.958	0.994	0.987	0.991	0.993	0.994	0.943	0.990	1.000	0.991	
R	0.995	0.987	0.998	0.997	0.997	0.996	0.997	0.979	0.992	0.991	1.000	

PCA of the GC fingerprints

The GC fingerprints of the rhizome of M. dauricum samples were then presented quantitatively by utilizing PCA. The relative peak area of seven distinctive chromatographic peaks (peaks 1, 2, 3, 4, 8, 9, and 10) in Fig. 2 was chosen in PCA. As depicted by the scree plot (Fig. 3A), the first three PCs had eigenvalues of 4.302, 1.088, and 1.015, explaining 61.45%, 15.55%, and 14.50% of variance correspondingly, representing 91.50% of the total variance. The following equations can be used to explain the principal components:

Figure 3 The scree plot (A) and plot of PCA scores (B) of tMDP’ GC fingerprints from different regions.

PC1 = 0.436 * X1 + 0.437 * X2 + 0.451 * X3 + 0.255 * X4 + 0.325 * X8 + 0.387 * X9 + 0.308 * X10

PC2 = −0.318 * X1 − 0.156 * X2 − 0.181 * X3 − 0.735 * X4 + 0.481 * X8 + 0.061 * X9 − 0.256 * X10

PC3 = 0.114 * X1 − 0.237 * X2 − 0.180 * X3 − 0.006 * X4 + 0.362 * X8 − 0.520 * X9 + 0.704 * X10

The PCA results were further analyzed by OriginPro 2021. As per the 3D scores plot of PCA depicted in Fig. 3B, all of the rhizomes of M. dauricum samples were integrated into a group within the 95% confidence ellipse, which indicated that the quality of the rhizome of M. dauricum from Northeast and Shandong of China was similar, which was consistent with the determination of polysaccharide content (Table 1). The high similarity suggested that the GC fingerprint was very suitable as a quality control indicator for the rhizome of M. dauricum. However, the rhizome of M. dauricum polysaccharides from different origins also differed slightly and could be clustered into three categories broadly, among which S1, S2, S4, and S8 were all produced in Shandong, S3, S5, S7, and S10 were from Heilongjiang and Jilin, while S6 and S9 were from Liaoning.

Immunomodulatory activity and mechanism of MDP

Effect of MDP on the morphology of RAW264.7 cells

tMDP was further purified by cellulose and gel columns to obtain a homogeneous polysaccharide fraction MDP. To examine the impact of MDP on RAW264.7 cells’ morphology, the morphologic changes of RAW264.7 cells were observed under a microscope. As could be observed in Fig. 4A, the cells that belonged to the control group had a round shape and normal structure. Nonetheless, following MDP treatment, there was a remarkable alteration in the structure of the cells. It is quite clear to observe that many of the antennas generated surrounding the cells may enhance the area of contact with exterior substances and were beneficial to the phagocytic uptake. This suggests that concentrations of MDP ranging from 10 to 400 μg/mL are capable of activating macrophages without causing cytotoxic effects.

Figure 4 Effect of MDP on the morphology of RAW264.7 cells (A) and the effect of MDP on RAW264.7 cell viability (B).

The values are presented as the mean ± SD, n = 3. **P < 0.01 vs. control group.

Effect of MDP on RAW264.7 cell viability

Macrophages are the primary cells implicated in the innate defense mechanism of the immune system. In addition, they are the major cells responsible for inflammation and other immunological processes that occur inside the host. It has been shown that many different polysaccharides stimulate both the activity of cells as well as their proliferation (Wang et al., 2018). As a result, we studied how MDP affected the viability of RAW264.7 cells, and the findings are depicted in Fig. 4B. At concentrations of 10, 50, 100, 200, and 400 µg/mL, the levels of cell viability for the various groups were 120.66%, 123.64%, 118.76%, 130.02%, and 114.73%, correspondingly, which illustrate that MDP has the potential to significantly improve RAW264.7 proliferation.

MDP enhanced immunomodulatory effects via TLR4 in vitro

Macrophages are by far the most important body cells that are associated with the immunological defense system (Cheng et al., 2019). After being activated, they are capable of producing a wide variety of chemokines and cytokines. One of the most important mechanisms that immunoregulators use is to cause the release of several biological components (IL-6, TNF-α, NO, etc.) (Tian et al., 2019). NO is a type of vital molecule that is produced by macrophages, and it played an important function in the modulation of apoptosis as well as the host’s defense against cancer cells and other pathogens (Ren et al., 2019). In addition, NO may also stimulate phagocytosis and lysis in macrophages, two processes that are essential to the immune system. Because of this, the ability of macrophages to release NO is an indication of the effects that polysaccharides have on the functioning of the immune system (Wang et al., 2017a). Inflammation, cancer, and immunological illnesses may all be traced back to the main active molecules inside organisms, such as TNF-α and IL-6, which perform integral roles in the pathogenesis of these conditions (Wang et al., 2017a). If the host is attacked by external pathogens, activated macrophages produce IL-6 and TNF-α for the purpose of mediating the immune system’s response (Sorimachi et al., 1999). Our previous study showed that MDP could promote the production of NO, IL-6 and TNF-α in RAW264.7 cells (Yang et al., 2022). These results suggested that MDP has significant immunomodulatory activities via the mechanism of increasing the release of NO, IL-6, and TNF-α in RAW264.7 cells. However, the mechanism of the in vitro immunomodulatory activity of MDP is unclear.

TLRs have been discovered as key membrane receptors that perform an instrumental function in activating macrophages (Fan et al., 2021). Among TLRs, several investigations have demonstrated that TLR4 is implicated in the generation of cytokines in response to polysaccharide stimulation (Liu et al., 2022b). To further investigate whether MDP exerts immune activity through activation of the TLR4 pathway, TAK-242, an inhibitor of TLR4, was utilized so that additional confirmation of the processes that result in MDP-mediated macrophage activation could be achieved. As depicted in Fig. 5, TAK-242 considerably decreased the levels of TNF-α, NO, and IL-6 (P < 0.05) as opposed to the MDP treatment group, illustrating that the TLR4 pathway was implicated in the MDP-mediated activation of macrophages.

Figure 5 Effects of MDP on the secretion of NO and cytokines with or without inhibitors.

(A) NO; (B) TNF-α; (C) IL-6. Without inhibitors, the secretion of NO and cytokines by macrophages was induced by MDP, however, this impact was hindered when the RAW264.7 cells had been pre-treated with inhibitors. The values are presented as the mean ± SD, n = 3. *P < 0.05, **P < 0.01, compared with Control group. #P < 0.05, ##P < 0.01, compared with the no inhibitor group.

MDP increased the protein expression of the TLR4-MAPK/NFκB signaling pathway

By performing a Western blot analysis, we evaluated how MDP affected the protein expression levels of TLR4 to get a deeper insight into its action mechanism. According to the results shown in Fig. 6A, MDP significantly promoted the expression of TLR4 protein, which is consistent with the above experimental results. Once TLR4 is activated, TLR4 is capable of binding with the MyD88 protein to create the TLR4-MyD88 complex, which then stimulates the phosphorylation of the downstream proteins, such as MAPKs (ERK, JNK, and p38) and NF-κB (Xin et al., 2019). The findings of Western blotting (Fig. 6B) illustrated that the protein expression of MyD88 was remarkably enhanced in RAW264.7 cells following MDP treatment, which indicated that TLR4 protein, after being activated by MDP, combined with a large amount of MyD88 and then activated the downstream pathway.

Figure 6 The influence of MDP on the stimulation of TLR4-MAPK/NFκB signaling pathways in RAW 264.7 cells.

(A) TLR4, (B) MyD88, (C) p-NFκB/NFκB, (D) p-ERK/ERK, (E) p-JNK/JNK, (F) p-P38/P38. The values are presented as the mean ± SD, n = 3. *P < 0.05, **P < 0.01, compared with Control group.

MAPKs perform crucial functions in immunological and inflammatory responses, including the stimulation of cell proliferation and the regulation of cytokine production via the modulation of transcriptional factors (Yang et al., 2019). Phosphorylation is a common mechanism that underlies the functions that are controlled by MAPKs (Yang et al., 2019). To determine if the MAPK signaling pathway was implicated in MDP-related macrophage activation, the phosphorylation levels of the p38, ERK, and JNK MAPK pathways were examined through western blot analysis. As depicted in Figs. 6D–6F, the phosphorylation levels of ERK, JNK, and p38 were substantially elevated in RAW264.7 cells following stimulation with MDP in contrast with the normal control cells. It has been shown that NFκB is a ubiquitous transcriptional factor that is essential for the activation of macrophages via the stimulation of diverse genes implicated in the modulation of immunological as well as inflammatory responses (Yang et al., 2019). As per the findings of Western blot (Fig. 6C), MDP caused an enhancement in the protein expression of p-NFκB in RAW264.7 cells. In conclusion, following MDP treatment, the key protein of TLR4-MAPK/NFκB signaling pathways including TLR4, MyD88, p- NFκB, p-ERK, p-P38, and p-JNK were up-modulated substantially in RAW264.7 cells. As per these findings, it is evident that MDP may stimulate the TLR4-MAPK/NFκB signaling pathways, which in turn exerts immunomodulatory impacts.

Discussion

The detection methods commonly used for monosaccharide composition include high-performance anion-exchange chromatography (HPAEC), gas chromatography (GC), and high-performance liquid chromatography (HPLC) (Liu, 2022a). Among them, the GC is an analytical method that is primarily utilized for characterizing and identifying volatile compounds and is an effective and sensitive technique employed to measure the composition of monosaccharides with the advantages of high sensitivity, good separation, fast analysis, and less sample consumption (Liu, 2022a). Cheong et al. (2016) used GC and HPSEC-RID-MALLS to compare and analyze the monosaccharide composition of three ginseng plant polysaccharides, Ginseng, American ginseng, and Panax notoginseng, which all consisted of six monosaccharides, but the intensity of each monosaccharide peak differed. This research can both ensure the quality of certain polysaccharides found in Panax spp., and perform a fundamental role in controlling the quality of the functional food products associated with these polysaccharides. Ding (2020) established a fingerprint profile of Lonicera japonica polysaccharide by GC, and the results of principal component analysis and factor analysis showed that L. japonica produced in Shandong had the best quality and those produced in Jilin were the last, indicating that different climatic environments could lead to differences in quality of L. japonica herbs from different origins. It serves as a baseline for determining the overall quality of herbs derived from L. japonica.

However, GC is not suitable for the determination of uronic acid fractions. Based on this, we determined the monosaccharide composition of tMDP by thin-layer chromatography. The results (Fig. S1) showed that tMDP did not contain uronic acid components, so GC could be used to detect its monosaccharide composition. Additionally, GC fingerprints may be established to offer a groundwork for the quality control of the rhizome of M. dauricum medicinal materials. A total of 11 common peaks were obtained by GC fingerprinting of tMDP, and 6 peaks were identified. Among them, peak 1 was Rha, peak 2 was Ara, peak 3 was Fuc, peak 8 was Man, peak 9 was Glu, peak 10 was Gal, and peak 11 was internal standard. However, peaks 5, 7, and 8 were also present in the mixed standards, which may be by-products of the derivatization process. Through GC fingerprinting and principal component analysis, the better quality herbs were screened out, thus laying the foundation for subsequent experiments.

Polysaccharides have been demonstrated to perform a broad range of biological functions, according to numerous studies, and the immunomodulatory activity of polysaccharides is widely acknowledged to be the most significant biological function of these molecules. Macrophages are immune cells that have a phagocytotic function to eliminate foreign antigens and perform an instrumental function in the case of antimicrobial invasion and host defense (Liu et al., 2022b). By coming into close touch with or eliminating infections or tumor cells, activated macrophages may generate cytokines that trigger other immune cells (Liu et al., 2022b). In this study, MDP was found to activate RAW264.7 cells and enhance the secretion of TNF-α, NO, and IL-6 indicating that MDP has immunomodulatory properties. Furthermore, the influence of MDP in promoting the release of TNF-α, NO, and IL-6 was suppressed after TAK-242 treatment. This suggests the possibility that MDP exerts immunomodulatory activity by activating the TLR4 protein pathway. The connection between the MAPK, TLR4, and TLR4 signaling pathways is known. When TLR4 is expressed in RAW264.7 cells, it activates two distinct signaling networks: the TRIF-dependent pathways and the MyD88-dependent pathways. Once the TLR4 signaling pathway has been activated, it stimulates the activation of the NF-κB and MAPK signaling pathways in a mechanism that is reliant on MyD88. The findings of western blotting illustrated that MDP could activate the expression of key node proteins on the TLR4-MAPK/NFκB pathway. In summary, the results of our research showed that MDP does, in fact, remarkably activate macrophages by activating TLR4, which is mediated by MAPK and NFκB signaling pathways (Fig. 7). This signaling pathway has also been shown to be responsible for the immunomodulatory activities of other polysaccharides, according to multiple research reports (Yang et al., 2019; Liu, 2022a; Wu et al., 2019).

Figure 7 Possible immunomodulatory signaling mechanism of MDP in RAW264.7 macrophages.

Conclusions

In this research, the extraction conditions of polysaccharides from the rhizome of M. dauricum were successfully optimized by RSM. The GC fingerprints of 10 batches of the rhizome of M. dauricum polysaccharides were established, and The findings of SA and PCA of the fingerprints illustrated that the rhizome of M. dauricum polysaccharides derived from various sources each had their own distinct chromatographic fingerprint features. Moreover, according to the findings of this research, MDP may substantially activate RAW264.7 cells through the TLR4-MAPK/NFκB dependent signaling pathway so as to generate biological cytokines. In conclusion, our studies have demonstrated an in vitro immunomodulatory effect of MDP with a molecular mechanism that is related to the activation of the TLR4-MAPK/NFκB signaling pathway.

Supplemental Information

Supplemental Information 1 TLC of the total polysaccharides from the rhizome of M. dauricum.

1. GlcUA; 2. Gal; 3. Glc; 4. Man; 5. The total polysaccharides from the rhizome of M. dauricum (tMDP); 6. Ara; 7. Fuc; 8. Xyl; 9. Rha; 10. Fru

Click here for additional data file.

Supplemental Information 2 Raw data.

The CDF files for Fig. 2 are the data of analyzing the fingerprints of 10 batches of the rhizome of M. dauricum Polysaccharide. The software to open this file is the Similarity Evaluation System for Chromatographic Fingerprint of Traditional Chinese Medicine, and the software installation package is available in the Figure 2 folder.

The OPJU file in the Figure 3 folder is the data of PCA analysis, and the open software is Origin, which can be downloaded and used for a fee from the Origin official website.

Click here for additional data file.

Abbreviations

GC Gas chromatography

RSM Response surface method

tMDP Total polysaccharides from the rhizome of M. dauricum

MDP Polysaccharides from the rhizome of M. dauricum

PCA Principal components analysis

TLR4 Toll-like receptor 4

MyD88 Myeloid differentiation factor 88 protein

MAPK Mitogen-activated protein kinase

NFκB Nuclear factor kappa-B

TCA Trichloroacetic acid

TFA Trifluoroacetic acid

MTT Methyl thiazolyl tetrazolium

LPS Lipopolysaccharide

RIPA Radioimmune precipitation assay

BCA Bicinchoninic acid

Additional Information and Declarations

Competing Interests

Author Contributions

Data Availability

We hereby declare that there are no conflicting interests involved in the present research. Li Li is employed by Sishui Siheyuan Culture and Tourism Development Company, Ltd.

Pei Yang conceived and designed the experiments, performed the experiments, analyzed the data, prepared figures and/or tables, and approved the final draft.

Yang Zhai conceived and designed the experiments, performed the experiments, analyzed the data, prepared figures and/or tables, and approved the final draft.

Yan Ma conceived and designed the experiments, performed the experiments, analyzed the data, prepared figures and/or tables, and approved the final draft.

Beibei Mao performed the experiments, prepared figures and/or tables, and approved the final draft.

Fengshan Wang analyzed the data, authored or reviewed drafts of the article, and approved the final draft.

Li Li analyzed the data, authored or reviewed drafts of the article, and approved the final draft.

Lijuan Luan conceived and designed the experiments, analyzed the data, authored or reviewed drafts of the article, and approved the final draft.

Yuhong Liu conceived and designed the experiments, analyzed the data, authored or reviewed drafts of the article, and approved the final draft.

The following information was supplied regarding data availability:

Raw data is available in the Supplemental Files.

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
