# Peer review of "Gas chromatography (GC) fingerprinting and immunomodulatory activity of polysaccharide from the rhizome of Menispermum dauricum DC"

_PeerJ, doi:10.7717/peerj.13946_

## Round 0.1 · original submission · Minor Revisions

Two experts assessed your manuscript and found it communicates relevant information that could be published after addressing the minor points raised, which are mainly related to the manuscript form and editing.

Reviewer 1 ·

Basic reporting

No comment

Experimental design

Experimental design is reasonable. The authors should include the number of replicates performed in each experimental design.

Validity of the findings

No comments

Additional comments

The manuscript is well written, and the data is clearly presented within the figures. The manuscript reports the optimisation of polysaccharides from MDP using GC analysis. The authors use PCA analysis to conclude that the polysaccharides from different sources had distinct chromatographic features. The manuscript is of general interest, and I propose the following minor comments for the authors to address before publication can be recommended.

1) The authors need to expand some abbreviations in the abstract for the benefit of the reader (e.g. PCA, MDP, TLR4)
2) Reference 1 line 46 (Anon 2009) cannot be verified and the authors should address this.
3) Line 83: The authors should explain what is meant by “Certain botanical polysaccharides have the potential to be detected”
4) Line 88: The statement “…plant products have been believed to be potential candidates for use as immunoregulators”. This statement would be strengthened by a supporting reference.
5) Line 230: Expand the abbreviations associated with Fig. 2C.
6) Line 232: The authors indicate that the seven different components identified in Fig. 2C could be identified from standards. Can the authors explain why some of the retention times are slightly shifted (e.g. peaks 9, 10, 11) in the standard vs tMDP.
7) Line 247: typo “there” should read “three”
8) Line 283: Can the authors provide an explanation for the improvement in cell viability in RAW264.7 cells?
9) Line 327: incorrect capitalisation of “The”.
10) Reference: Yu et al (J. Log Uni PAP) – this reference cannot be identified

Reviewer 2 ·

Basic reporting

In general, the investigation is carried out in a good way.
The approaches and experiments to elucidate the possible role of M. dauricum polysaccharides as immunomodulators are correct.

Experimental design

The experiments are carried out with scientific rigor, however, some of the methods mentioned are lacking references, later I will detail which ones I am referring to.

Validity of the findings

The results show the possible role of polysaccharides as modulators of the immune response, however, the conclusions that emerge from the research should be taken with more reservation; This is due to the fact that the tests are carried out in vitro and no animal model has been used that shows the same behavior that leads to proposing them as potential medicinal agents or in the food industry.

Additional comments

The following corrections and suggestions are for the improvement of the manuscript.

line 24: Full name of M.dauricum.

line 29: Full name of PCA analysis.

lines 77-80: I suggest you slightly change what you write in these lines. Maybe: In studies to date, there are no reports indicating that these products have toxic effects.

line 89: After this line, you could add information about the lack or no information about the possible triggering of allergies or adverse effects from the use of said compounds.

line 94: Delete the word "successfully".

Table 1: In the legend, the full name of (PCA and tMDPs)

line 106: full name of LPS.

line 111: full name of RSM

In the abstract or introduction section, mention what GC fingerprinting is (as we found this until the discussion section)

line 128: full name of TCA

lines 131-135: Reference requiered

lines 137-143: ibidem

line 159: full name of MTT

lines 158-171: Reference

In the legend of Fig 2, delete the word "extremely"

Place the number bar in Fig. 4

Lines 285-286: reference
Lines 287-289: ibidem
Lines 289-291: ibidem
Lines 291-294: ibidem
Lines 294-296: ibidem
Lines 296-298: ibidem
Lines 303-342: ibidem
Lines 340-342: ibidem
Lines 371-373: ibidem

Lines 353, 355 and 357: L.japonica

Line 189: Mention the concentration of the protein placed on the SDS-PAGE.

The Fig. S1: I think it is important that you put a figure caption with legends that allow a quick analysis of the chromatography and the absence of the unwanted component.

---

## Round 0.2 · accepted · Accept

The manuscript was improved following the Reviewers' comments. As a consequence, it is now suitable for publication in PeerJ.